# Structural and Micromechanical Properties of Ternary Granular Packings: Effect of Particle Size Ratio and Number Fraction of Particle Size Classes

**DOI:** 10.3390/ma13020339

**Published:** 2020-01-11

**Authors:** Joanna Wiącek, Mateusz Stasiak, Jalal Kafashan

**Affiliations:** 1Institute of Agrophysics, Polish Academy of Sciences, Doświadczalna 4, 20-290 Lublin 27, Poland; mstasiak@ipan.lublin.pl; 2Department of Mechanical Engineering in Agro-Machinery & Mechanization, Agricultural Engineering Research Institute (AERI), Agricultural Research Education and Extension Organization (AREEO), 31359-33151 Karaj, Iran; KAFASHAN@engineer.com

**Keywords:** discrete element method, ternary granular packings, structure, micromechanics, particle size ratio, number fraction of particle size classes

## Abstract

The confined uniaxial tests of packings with discrete particle size distribution (PSD) were modeled with the discrete element method. Ternary packings of spheres with PSD uniform or nonuniform by number of particles were examined in three-dimensional (3D) system. The study addressed an effect of the particle size ratio and the particle size fraction on structural and micromechanical properties of mixtures. A study of packing structure included porosity and coordination numbers, while the investigation of micromechanical properties included distribution of normal contact forces and stress transmission through the packing. A micro-scale investigation of the effect of particle size ratio on structure and mechanics of the ternary packings revealed a strong relationship between the properties of sample and the value of parameter till its critical value was reached. A further increase in particle size ratio did not significantly affect properties of packings. Contrary to the porosity and coordination numbers, the partial stresses were highly affected by the fraction of particle size classes in ternary mixtures. The contribution of the partial stress into the global stress was determined by number fraction of particles in packings with small particle size ratio, while it was mainly determined by particle size ratio in packings with small particle size ratio.

## 1. Introduction

Ternary mixtures, after binary ones, are the simplest polydisperse granular packings, which exhibit interesting and unexplained behavior. An examination of their properties is very important and necessary because the particulate systems composed of three particle size fractions are commonly used in many branches of industry (e.g., chemistry, pharmacy, and metallurgy).

In the past, extensive efforts have been made to investigate the structural properties of granular packings that are composed of non-uniformly sized particles by using experimental [1,2,3,4] and numerical methods [4,5,6,7,8]. A number of these studies has been devoted to the analysis of properties of ternary granular mixtures, however, they mainly included an analysis of the packing density and the coordination number, the most important parameters in describing the geometrical arrangement of particles in grain bedding [1,9,10,11,12]

An experimental study on ternary mixtures of approximately spherical particles, which was conducted by Westman and Hugill [9], showed that the apparent volumes of mixtures containing unit real volume of solid fall between limiting values that can be calculated from simple assumptions, and that their deviation from these limits depends in a definite manner upon the diameter ratios of the component particles. In 1961, experimental and theoretical studies on density of ternary packings of nearly spherical steel shots, as conducted by McGeary [1], extended the study of Westman and Hugill [9]. A number of attempts have been made for the next few years to model mathematically the packing density of multi-sized granular assemblies [10,11,12]. In 1986, Stovall et al. [11] proposed a linear density model of multicomponent systems, which was applicable inter alia for binary and ternary mixtures. It assumed that the specific volume of the mixture was a linear function of the volumetric fractions of the individual size classes. Deviations from this theory at certain volumetric fractions, which lead to decrease in packing density, and result in a non-linear particle packing model, as developed by Kwan et al. [13] and Wong and Kwan [10] for binary and ternary mixtures, respectively. In 2018, Yan et al. [12] proposed a mathematical model for calculating the primary porosity of unconsolidated sands based on the grain size distribution and packing texture. The authors proved that the new method is applicable for binary and ternary mixtures of sand.

Two-dimensional image modelling of binary and ternary granular packings, as performed by Mota et al. [14], have shown a strong relationship between the ratio of the diameter of the largest and the smallest particle in system and the volume fraction on the porosity and tortuosity of granular system. Numerical study on the structural properties of granular assemblies with particle size distributions uniform by number of particles, divided into three, five and seven particle size classes, as conducted by Wiącek et al. [15], by using a discrete element method (DEM), revealed that the packing density of ternary samples increased when the particle size ratio increased. The opposite tendency was observed for mean overall coordination number, which decreased with increasing particle size ratio. An experimental study on the effect of the volume fraction of the components on the coordination number in ternary mixtures of spheres, which were carried out in 2003 by Zou et al. [3], indicated that the overall coordination number remained independent of volume fraction of particle size classes. However, the authors observed that the partial coordination numbers varied with volume fraction of components. These findings were later confirmed by the numerical results that were obtained by Yi et al. [16]. Based on the DEM simulated results, the authors proposed a mathematical model to describe the distributions of partial coordination number in ternary granular packings.

The composition of multi-sized mixtures, determining packing density and distribution of contacts in granular assemblies, also has great influence on their mechanical properties. Micromechanical properties (e.g., distribution of normal contact forces, stress transmission, mobilisation of friction at contact points) and macromechanical properties (e.g., flowability, permeability, elasticity and strength) of granular packings were both found to be affected by degree of particle size heterogeneity and volume fraction of particle size classes [17,18,19,20,21].

Martin and Bouvard [17] observed that the stiffness of binary packings subjected to isostatic compaction increased with an increase in the number of large particles and the particle size ratio. Numerical simulations of compression of polydisperse packings, as performed by Gu and Yang [22], have shown a decrease in the modulus of elasticity with an increase in the particle size heterogeneity that was confirmed by Wiącek and Molenda [23] for polydisperse mixtures with normal particle size distribution (PSD). Later studies conducted by Wiącek et al. [21] for binary mixtures of spheres have shown a significant effect of both, the ratio of the diameters of the smallest and the largest particles and the number fraction of particle size classes on the mechanical properties of packings. The distributions of normal contact forces and the partial and global stresses varied with an increase in particle size ratio and content of small particles in mixture. A decrease in the energy dissipated in granular bedding with an increase in the contribution of small spheres in system and with a decrease in ratio of the diameters of the smallest and the largest particles was observed. Numerical simulations of the compression of polydisperse packings, as prepared by Göncü et al. [19] and Wiącek et al. [21], revealed higher pressure in the samples with greater particle size heterogeneity. Wiącek et al. [21] found that the global stress and packing density followed the same path with increasing contribution of small spheres in binary mixtures. Iddir et al. also reported a strong relationship between the particle size ratio and the contribution of particle size classes and the stresses in binary and ternary particulate systems in simple shear flow [18].

The study of the relationship between the particle size distribution and the structural and mechanical properties of grain assemblies is of great importance to the university research and the industry because both natural granular materials and the ones involved in industrial processes are nonuniform in size. Over the last few decades, a number of studies have been conducted to investigate the relationship between the structural properties and the mechanical behavior of particulate assemblies with the particle size heterogeneity. Although few investigations were devoted to ternary mixtures with various degree of polydispersity and various volume fractions of particle size classes, a review of the literature shows that these studies mainly focused on the analysis of packing density and coordination number of mixtures. A detailed studies on the micromechanical properties of ternary granular packings with PSD uniform and nonuniform by number of particles, including a contribution of the mechanically stable particles or particles of various sizes into the force transmission through the system, the relationship between the partial coordination number and particle size ratio or distribution of contact forces in the samples with various particle size ratios and number fractions of particle size classes, are very scarce and insufficient. Knowledge in that field seems to be the most demanded in the branches of industry, wherein polydisperse granular materials are subjected to external loads during such processes as tableting, pressing, agglomeration et al. These are, inter alia, pharmaceutical, metallurgical, ceramic, food, and energy industries. Granular materials are subjected to various types of loading conditions during technological processes that frequently result in products of properties different than the properties of components. A production of the stable and uniform products of high quality and the establishment of the optimal handling, storage, and conveying conditions for these materials requires good knowledge of their structural and micromechanical properties. Many of these solids exhibit difficult handling behaviors, which gives rise to considerable challenges in the design and operation of the handling and processing.

An increase in the demand for the knowledge of the micromechanical properties of polydisperse granular packings, which results from increasing interest in granular materials and their application in many branches of industry makes that issue very important and worthy of further investigation.

Therefore, this study focuses on the micro-scale analysis of the effect of particle size ratio and the contribution of particle size fraction on structural and micromechanical properties of mixtures composed of three grain size classes. A discrete element method DEM [24] was applied to model a confined uniaxial compression test of granular packings, which is one of the methods most frequently recommended for measuring mechanical parameters of grain beddings. A study on ternary granular materials with PSDs uniform and nonuniform by number of particles was conducted, providing a significant knowledge on these idealized sets of spheres. Knowledge gained from presented project may lead to better understanding of real granular mixtures with more complex PSDs and might find application in industries dealing with granular materials.

## 2. Materials and Methods

The three-dimensional simulations were conducted using the EDEM software (version 2.7) [24], based on the Discrete Element Method [25]. A simplified viscous-elastic non-linear Hertz–Mindlin contact model was used [26], wherein the normal contact force ($F_{ij}^{n}$), which results from the contact of particle *i* with particle *j*, is expressed as:(1)$$F_{ij}^{n}=k_{n}\delta_{n}^{\frac{3}{2}}-2\sqrt{\frac{5}{4}}\beta \sqrt{k_{n}m^{*}}V_{n}^{rel}\delta_{n}^{\frac{1}{4}},$$
The tangential force is obtained from:(2)$$F_{ij}^{t}=-F_{ij}^{t,n}-2\sqrt{\frac{5}{6}}\beta \sqrt{k_{t}m^{*}}V_{t}^{rel},$$
where
(3)$$F_{ij}^{t,n}=F_{ij}^{t,n-1}+k_{t}\Delta \delta_{t}.$$

In Equations (1)–(3), *k_n_* and *k_t_* are the normal and tangential stiffness coefficients, *δ_n_* and *δ_t_* are the normal and cumulative shear displacements, *m** is an equivalent mass, and $V_{n}^{rel}$ and $V_{t}^{rel}$ are the normal and tangential components of the relative velocity. In Equation (2), Δ*δ* is the relative tangential displacement of the contacting surfaces occurring in the *n*-th timestep. The stiffness coefficients may be expressed as:(4)$$k_{n}=\frac{4}{3}Y^{*}\sqrt{R^{*}},$$
(5)$$k_{t}=8G^{*}\sqrt{R^{*}\delta_{n}},$$
where $Y^{*}$ is an equivalent Young’s modulus; $R^{*}$ is an effective radius of contacting particles; and $G^{*}$ is an equivalent shear modulus. The elastic constants, Young’s modulus *Y*, and shear modulus *G*, are related to each other, as follows:(6)$$G=\frac{E}{2\left(1+\nu \right)},$$
where *ν* is a Poisson’s ratio.

The coefficient of restitution (*e*) dependent coefficient *β* in Equations (1) and (3) is given by:(7)$$\beta =\frac{\ln e}{\sqrt{\ln^{2}e+\pi^{2}}}.$$
The time step is set to small to allow for the assumption of constant translational and rotational accelerations. The motion of each particle is given by:(8)$$m_{i}\frac{dV_{i}}{dt}=\sum_{j}\left(F_{ij}^{n}+F_{ij}^{t}\right)+m_{i}g,$$
(9)$$I_{i}\frac{d\omega_{i}}{dt}=\sum_{j}\left(R_{i}\times F_{ij}^{t}\right)+\tau_{rij},\;i=1,\;2$$
where *m_i_*, *R_i_*, *I_i_*, *V_i_*, and *ω_i_* are, respectively, the mass, radius, moment of inertia, linear velocity, and angular velocity of particle *i*; *g* is acceleration due to gravity, which is assumed to be constant. The normal and tangential forces result from the contact of particle *i* with particle *j*. Integrating Equations (8) and (9) gives the velocity and position of the particles. The torque of particle, related to rolling friction *µ_r_*, is expressed, as follows:(10)$$\tau_{rij}=-\mu_{r}F_{ij}^{n}l_{i}\omega_{i},$$
with *l_i_* being a distance of the contact point from the center of mass of particle *i*.

Tangential contact force was limited by the Coulomb friction law assuming that particles slide over each other when the tangential force is at limiting friction. The Coulomb friction law is expressed as:(11)$$F_{ij}^{t}<\mu_{s}\left|F_{ij}^{n}\right|,$$
where *μ_s_* is the coefficient of static friction.

The rigid particles were allowed to locally overlap at contact points while using a soft contact approach. A conservation of volume of bedding was assumed since the overlaps between particles did not exceed 0.01% of particle diameters. Table 1 lists the input parameters for DEM simulations, corresponding to the mechanical parameters of steel rods and steel walls.

Numerical tests of the confined uniaxial compression of granular packings was performed, which is a standard laboratory test procedure to measure the mechanical properties of granular materials [27]. The simulations contained three stages: sample generation, compression test, and relaxation of granular packing. In the sample generation stage, spheres with random initial coordinates were generated inside the box with rectangular cross-section of 0.12 m × 0.1 m and 0.12 m thick, which were placed above a chamber of uniaxial compression apparatus of the same size (see Figure 1). The size of the sample, regarded by authors as a representative elementary volume [28], allowed for neglecting the wall effects. The spheres settled down onto the bottom of the test chamber under gravity in a dispersed stream, so-called rain filling. In the second stage of simulation, spheres were compressed through the top cover of the chamber that moved vertically downwards at a constant velocity of 3 m/min until a maximum vertical pressure on the uppermost particles reached 100 kPa. In the last stage of simulations, a relaxation of compressed granular packings was conducted until a total kinetic energy of particles reached 10^−6^ J. The rigid and frictional walls of apparatus were modelled, which did not deform under the applied load. The test procedure followed the recommendations of the Eurocode 1 [27]. Three replicate tests were run for each sample to verify the repeatability.

The simulations were carried out while using samples composed of three particle size fractions. The samples were described by particle size ratio *g*, defined as a ratio between the diameter of the largest and the smallest spheres. Mixtures with particle size ratio varying from 1.25 to 5 were generated in simulations. Each sample consisted granulometric fraction composed of spheres with diameter of 6 mm and two fractions with smaller particles. The differences between the diameters of particles in neighboring size fractions were equal. The samples had particle size distribution uniform by number of particles (i.e., an equal number of particles of different sizes) or nonuniform by number of particles (i.e., a different number of particles of different sizes). Mixtures with PSD nonuniform by number of particles were described by number fraction of particle size classes *f*, which defined a percentage content of spheres representing different particle size classes to total number of particles in mixture. Table 2 presents the total number (*N*) and diameters of particles (*D_k_*) in mixtures with uniform PSD, while number fraction of particle size classes (*f_k_*), number of particles in fractions (*N_k_*), and total number of particles (*N*) in mixtures with uniform PSD and *g* = 1.25 and 3.75 are shown in Table 3.

## 3. Results

The micro-scale analysis of the effect of the particle size ratio and number fraction of particle size classes on the structural and mechanical properties of ternary mixtures with discrete PSD included porosity, compression index, number of contacts, distribution of contact forces, and transmission of stress through the sample.

### 3.1. Effect of the Particle Size Ratio

#### 3.1.1. Porosity and Compression Index

Figure 2 presents the evolution of the porosity of samples (*Φ*) with different values of *g*, which were subjected to compressive load. The mean values are shown; the error bars indicate ±one s.d. At compressive pressures smaller than 15 kPa, initial fast consolidation of loose material took place, resulting in a sharp decrease in the porosity. At greater compression pressures, the rearrangement of particles in consolidated sample was limited and the porosity decreased weakly. An increase in value of *g* from 1.25 to 2.5 resulted in a substantial decrease in porosity that was not observed for larger *g* values. In the entire range of compressive load, the differences between porosities of mixture with *g* ≥ 2.5 fell within the range of scatter. A rate of decrease in porosity of the mixtures decreased with an increase in *g* value, and the *Φ* value remained almost unchanged when *g* > 2.5. These results indicated the presence of a critical particle size ratio above which porosity of ternary mixtures remained unchanged. The exceedance of *g* = 2.5 did not significantly change the volume of empty space between large particles providing approximate porosities. These results corroborate the findings of experimental and numerical studies that were conducted by Furnas [29] and Wiącek et al. [13], respectively, who examined mixtures composed of two, three, and more particle size classes. The authors reported that when critical value of particle size ratio was exceeded, different for samples with various number of particle size fractions, density of packings has only been changed slightly. The experimental studies prepared by Mc’Geary [1] and Shire et al. [30], and numerical investigations that were conducted by Shire et al. [30] and Sánchez et al. [7] for binary granular mixtures confirmed the findings that were presented by Furnas [29].

A compression index (*C*), which is the important compressibility parameter adopted from a soil mechanics [31], was determined for each sample. The compression index, being a measure of change of sample volume resulting from a change in loading, was defined, as follows:(12)$$C=\frac{\Delta \Phi }{\Delta \log\left(\sigma_{z}\right)},$$
where Δ*Φ* is a variation of the porosity and Δlog(*σ_z_*) is a variation of logarithm of vertical load. Figure 3a shows a *Φ*(log(*σ_z_*)) relationship for sample with *g* = 2.5. The compression index was determined for linear part of *Φ*(log(*σ_z_*)) curve (see inset of Figure 3a). Figure 3b presents the relationship between the compression index and the particle size ratio for ternary packings with different values of *g*. In spite of the largest porosity, granular packings with *g* = 1.25 had small compressibility as compared to assemblies with *g* = 2.5. Nearly ordered structures that formed by similarly sized spheres exhibited smaller ability of particles to rearrange within the sample providing smaller compression index. Increase in *C* value of 23% was observed with an increase in the value of *g* from 1.25 to 2.5, in spite of the decreasing porosity of sample. The evolution of the packing structure of samples from nearly ordered to more disordered allowed particles to rearrange within the sample, increasing its compressibility. A further increase in *g* resulted in a decrease in *C* value. Compression index of packings decreased by above 45% when the particle size ratio increased from 2.5 to 5, despite similar porosities of mixtures. Bodman and Constantin [32], Lade et al. [33] and Sun et al. [34] reported earlier that the compressibility of soil was affected by both, porosity, and granulometric composition of sample. In this study, no evident relationship between porosity of packing and compression index was observed for the system of idealized ternary mixtures composed of ideal spheres. It was found that the compressibility of granular material was mainly determined by the degree of the order in the packing for particle size ratios not larger than 2.5. The compression index was strongly influenced by the *g* value in the case of more heterogeneous samples.

#### 3.1.2. Coordination Numbers

A degree of filling of empty spaces between particles in granular mixtures determines the number of contacts in sample. The relationship between an average coordination number (*CN*), being defined as a ratio between the number of interparticle contacts and the number of particles in system, and Figure 4 presents the particle size ratio. Coordination numbers were calculated for representative elementary volumes of mixtures, being placed in the central part of the sample, in order to reduce the effect of wall on value of parameter. Nearly 3% decrease in average coordination number was observed that was due to the evolution of the packing structure of samples from nearly ordered to more disordered with less number of interparticle contacts. Despite a decrease in the porosity of the packing with an increase in *g* value from 1.25 to 2.5. A further increase in particle size ratio did not affect significant CN value, which remained similar for samples with *g* ≥ 2.5. In granular packings with *g* = 1.25, nearly each particle was supported by neighboring particles and it supported other ones. In contrast, in more polydisperse mixtures, a certain number of particles did not support neighboring particles, which decreases the average coordination number. These results corroborate findings of Shire et al. [30] and Wiącek and Molenda [23] for binary packings and mixtures with normal *PSD*, respectively. The authors observed smaller number of contacts in samples with greater degree of particle size heterogeneity.

The evolution of a corrected coordination number (*CN**) (including mechanically stable particles) with the particle size ratio is also shown in Figure 4 since only mechanically stable particles with more than three contacts contribute to force transmission through the system [18,35]. The corrected coordination number increased in the entire range of *g* value. An increase in value of *g* from 1.25 to 5 resulted in an increase in *CN** by 12%. Higher values of corrected the coordination number in mixtures with larger particle size ratios were due to an increase in the number of particles with more than ten contacts (see Figure 6). These results corroborate the findings of Wiącek [4], who observed an increase in corrected coordination number with increasing *g* value in the binary granular packings.

Figure 5 presents the percentage of particles with number of contacts (*N_c_*) smaller than 4 (termed non-rattlers) and particles with *N_c_* ≥ 4 (termed rattlers) in the mixtures with different value of *g*. In all of the packings, the percentage of non-rattlers was the smallest among the particles with the largest diameter *D*_1_ = 6 mm (Figure 5a). The largest percentage of non-rattlers was observed among the particles with the smallest diameters *D*_3_. In mixtures with *g* = 1.25, percentage of non-rattlers among particles with large and medium size (*D*_1_ and *D*_2_) was larger when compared to the packings with greater values of *g*. In the case of the smallest spheres, the percentage of non-rattlers was the largest in samples with *g* > 1.25. No effect of the sizes of medium and small particles on the percentage of non-rattlers in mixtures with *g* > 1.25 was observed, which indicated no relationship between the percentage of mechanically unstable particles and the particle size ratio in these ternary mixtures. The percentage of rattlers among large and medium particles increased with an increase in *g* value, contrary to rattler percentage among small spheres (Figure 5b). According to expectations, in samples with a given value of *g*, the rattler percentage among large spheres was greater as compared to medium particles. The smallest differences between the rattler percentage among particles with various diameters were obtained for samples with *g* = 1.25 that increased with an increase in the value of *g*. In granular packings with small *g*, ordered structures were formed by similarly sized particles providing a more homogeneous distribution of number of contacts among spheres with various sizes (Figure 6). More heterogeneous distribution of number of contacts among spheres with various sizes was observed because the polydispersity prevented the formation of a more ordered structure, in mixtures with larger *g*.

Figure 6 shows the probability distributions for the number of contacts per particle in mixtures with different particle size ratios. In samples with *g* = 1.25, a peak was observed at *n* = 7; however, large number of particles had also 6 and 8 contacts. A peak value decreased when *g* increased. In packings with *g* > 2.5, the peak value tended to 3 and fraction of spheres with *n* = 3 increased with an increase in the value of *g*. These results corroborate the findings of Taiebat et al. [36], who observed a decrease in the peak value with an increase in the value of *g* in polydisperse samples with PSD being uniform by volume fraction.

A detailed study regarding the contribution of small (*s*), medium (*m*) and large (*l*) particles into the number of contacts in the examined mixtures was conducted through comparison of the percentage of number of contacts between small spheres (*ss*), small and medium spheres (*sm*), small and large spheres (*sl*), medium spheres (*mm*), medium and large spheres (*ml*), and large spheres (*ll*) (Figure 7). A fraction of contacts between small particles decreased above twofold with *g* value increasing from 1.25 to 2.5. A further increase in the particle size ratio to 3.75 decreased the fraction of *ss* contacts by next few percentage. A negligible difference between fraction of *ss* contacts was observed in mixtures with the largest *g* values. For contacts between small and medium particles, relationship between fraction of contact and *g* value was similar to the one that was obtained for *ss* contacts. In the case of *sl* contacts, their contribution into the number of all contacts decreased gradually with increasing *g* value. The same tendency was observed for contacts between medium spheres. The contribution of contacts between medium and large particles and between large particles was found to increase with increasing *g* value. It is worth noting that, in mixtures with *g* = 3.75 and *g* = 5, the differences between contributions of the respective contacts into the total contact number were the smallest, and they mostly lied within the range of scatter. These results confirm the presence of a critical particle size ratio in ternary mixtures, above which partial contact numbers remain almost unchanged.

#### 3.1.3. Contact Forces and Stress Transmission through the Contacts

Figure 8a illustrates the probability distribution functions of normal contact forces (*F_n_*) in the examined samples. The probability distributions were more asymmetrical and narrow for packings with larger values of *g*. The dominance value of *F_n_* and an average contact force (*F_n aver_*) both decreased by above 20% (Figure 8b) when value of *g* increased from 1.25 to 2.5. An increase in value of *g* from 2.5 to 5 both slightly changed the distribution of normal contact forces and the dominance value of *F_n_*. These results corroborate the previous findings of Wiącek et al. [21], who observed that for binary packings of spheres *f*(*F_n_*) curves narrowed and value of *F_n_* decreased with increasing difference between the particles’ sizes. Wiącek and Molenda [23] observed the opposite effect w for polydisperse granular packings with normal PSD, for which the probability distribution functions became broader with an increase in the degree of particle size heterogeneity. This inconsistency might be due to the different PSDs in the examined samples that strongly determine micromechanical properties of granular packings [6].

An effect of the particle size ratio on the distribution of contact forces in ternary mixtures has been studied because the contact forces in polydisperse granular packings were found to be strongly determined by the size of particles in system [5,36,37]. The particles were divided into two groups: first group included spheres that transmitted at least one strong force, exceeding 75% of *F_n aver_* (termed involved particles), and the other group of particles transmitted weaker forces (termed uninvolved particles). Figure 9 presents the percentage of particles involved in the strong force transmission in the mixtures with different values of *g*. In all of the packings, the percentage of spheres involved in the strong force transmission was the largest among particles with *D*_1_ = 6 mm and increased with an increase in value of *g*. The percentage of involved particles among medium spheres decreased when value of *g* increased from 1.25 to 2.5, which was not observed for larger *g* values. Among the smallest spheres, the percentage of particles that were involved in the strong force transmission was the largest in mixtures with *g* = 1.25 and decreased with an increase in value of *g*. The differences between percentage of involved particles with various sizes was the smallest in the packings with *g* = 1.25, where the contact force network was the most homogeneous. The packings became more heterogeneous with an increase in value of *g*. These results are in agreement with the findings of Voivret et al. [5] and Wiącek and Molenda [35], who reported the dominant role of the largest particles in the transmission of contact forces in polydisperse mixtures, and an increase in the number of small particles excluded from the force network with an increase in the degree of particle size polydispersity.

The effect of the particle size ratio on the global and partial stresses in ternary granular packings was studied. The mean particle stress is defined as [38]:(13)$$p^{p}=\frac{1}{3}tr(\sigma^{p})$$
where the stress tensor components for a single particle are given by [19]:(14)$$\sigma_{ij}^{p}=\frac{1}{V_{p}}\sum_{c=1}^{N_{c}}l_{i}^{pc}F_{ij}^{n\;pc}.$$
(15)$$\sigma_{ij}^{p}=\frac{1}{V_{p}}\sum_{c=1}^{N_{c}}l_{i}^{pc}F_{ij}^{n\;pc}.$$

In Equation (14), *V_p_* is a particle volume, $F_{ij}^{\;n\;pc}$ is a normal force exerted on particle *p* at contact *c,* and *N_c_* is a number of contacts of particle *p*. The branch vector connecting the centre of the particle to its contact ($l_{i}^{pc}$), as associated with particle radius $R_{i}^{p}$, displacement in normal direction $\delta_{n}^{c}$ and contact-direction unit vector $\hat{n}$ is expressed as:(16)$$l_{i}^{pc}=\left(R_{i}^{p}-\frac{\delta_{n}^{c}}{2}\right)\hat{n}.$$

The mean normal stress for the whole sample comprising *N* particles (global stress) is given by:(17)$$p=\frac{1}{V}\sum_{p=1}^{N}\left(p^{p}V^{p}\right)$$
where *V* is a volume of sample.

For ternary mixtures, three types of the partial stresses are distinguished: transmitted by small particles (*p_small_*), medium particles (*p_medium_*) and large particles (*p_large_*). Figure 10 shows the evolution of the global and partial stresses with the particle size ratio in mixtures being subjected to compressive load of 100 kPa. The smallest global stress was observed in the packings with *g* = 1.25, which increased by above 8% and 50% in mixtures with *g* value of 2.5 and 3.75, respectively. Contrary to expectations, a further increase in *g* value to 5 resulted in 5% decrease in global stress when compared to one that was calculated for *g* = 3.75. In all of the packings, the contribution of the partial stress to the global stress decreased with decreasing particle size. That effect was the weakest in samples with *g* = 1.25, where the contact force network was the most homogeneous. The stress transmitted by large spheres increased by 73% with an increase in the *g* value from 1.25 to 2.5, and by next 90% when particle size ratio increased to 3.75 and 5. The opposite tendency was observed for pressures that were transmitted by medium and small particles that decreased with an increase in *g* value.

A slight difference between pressures in samples with *g* = 3.75 and *g* = 5 indicated the presence of a critical value of particle size ratio above which further increase in *g* value did not affect significantly transmission of pressures through the contact network in ternary mixtures with particle size distribution uniform by a number of particles.

### 3.2. Effect of the Number Fraction of Particle Size Classes

#### 3.2.1. Porosity and Compression Index

Figure 11a shows the porosities of packings with various number fraction of particle size classes for *g* = 1.25 and *g* = 3.75. Regardless of the *g* value, the porosity increased by few percentage with an increase in the number of large spheres. The largest porosities were obtained for mixtures that were composed of the same number of large and medium spheres. Wiącek [4] also observed a decrease in the porosity with a decrease in the contribution of small particles in mixtures w in binary samples, provided that the volume fraction of small spheres did not exceed 60%. The negligible differences between the compression indexes calculated for samples with various value of *f* (Figure 11b) indicated no effect of the content of particles with different sizes on *C* value.

#### 3.2.2. Coordination Numbers

Figure 12 shows the average coordination numbers and the corrected coordination numbers in ternary mixtures with different number fraction of particle size classes. In the case of the samples with *g* = 1.25, in which the differences between particle diameters are relatively small, no evident effect of particle content on value of *CN* and *CN** was observed. The differences between the values of *CN* calculated for various samples lied within the range of scatter. The probability distributions of number of contacts per particle were found almost identical in these packings, regardless of the contribution of the individual particle size classes in samples (Figure 13a). In mixtures with *g* = 3.75 and different *f* values, average coordination numbers differed by few percentage. These results corroborate the findings of Zou et al. [3], who reported that, for ternary granular assemblies, the number of contacts between particles was governed by both, statistical and geometric factors. The authors indicated that the number of contacts involving a given component increased with an increase in the volume fraction of that component in sample, and large particles had greater number of contacts because of their large surface area. In mixtures with *g* = 3.75, a tendency in the variation of *CN** value with a change in the content of individual particle classes was opposite when compared to *CN*. Corrected coordination number increased by few percentage in packings with *f* = 424, as compared to samples with *f* = 244. The value of *CN** was the smallest in mixtures with *f* = 442, which was due to the smallest number of spheres with *N_c_* ≥ 12 (Figure 13b).

#### 3.2.3. Stress Transmission through the Contacts

Figure 14 shows the global and partial stresses in the mixtures with different fractions of particle size classes for *g* = 1.25 and *g* = 3.75. Regardless of the *g* value, the smallest global stress was observed in mixtures that were composed of the same number of medium and small spheres (*f* = 244), which increased in packings with *f* = 424 and *f* = 442. Although the global stresses in the latter ones were similar, the different partial stresses in these samples were observed. For *g* = 1.25, in packings with the same number of medium and small spheres, the largest stress was obtained for medium particles, while the stress calculated for small spheres was 6% higher than the one that was obtained for large particles. In spite of two times smaller number of large particles as compared to small and medium ones, large spheres played an important role in transmission of stress through the medium. In samples with *f* = 424, the largest stress was obtained for large particles that decreased for small and medium spheres, respectively. According to the expectations, in packings with *f* = 442, large particles transmitted the largest stress, while the smallest stress was obtained for small spheres. In mixtures with *g* = 3.75 and *f* = 244, the least numerous large particles transmitted the largest stress that decreased with a decrease in particle size. These results indicate a huge significance of the difference between sizes of particles on contribution of partial stress into a global stress in ternary granular packings. For mixtures with *f* = 424 and *f* = 442, the largest stress was obtained for the large spheres, while the smallest partial stress was obtained for small particles. Despite the two times smaller number of medium particles in packings as compared to small particles, the contribution of the former ones into the global stress was two times larger. These results show that the contribution of the partial stress into the global stress is affected by both number fraction of particle size classes and particle size ratio. While, in mixtures with small *g* value, particle content determines contribution of the partial stress into the global stress, in packings with larger *g* values, the contribution of the partial stress into the global one is determined by particle size ratio.

## 4. Conclusions

Wiącek et al.previously conducted the detailed studies on the effect of the particle size heterogeneity and the content of particle size classes on the structural and micromechanical properties of binary granular packings [4,21]. Because ternary particulate systems are commonly used in many branches of industry and the knowledge of some of their properties remains scarce, the previous study has been extended to ternary mixtures. Although, in some cases, the binary and ternary granular mixtures exhibit similar behaviors, the latter ones exhibit certain characteristics, typical of these materials. Therefore, in this study, the relationship between the particle size ratio and the number fraction of particle size classes in ternary granular packings, and their structural and mechanical properties were investigated in the micro-scale. A confined uniaxial compression of mixtures that were composed of three particle size fractions was simulated with a discrete element method.

In the last few decades, various efforts have been made to study the properties of ternary packings, however, they were mainly devoted to beddings with different volume fractions of the components. In this study, packings with PSD uniform by number of particles (with equal number of spheres of each diameter), or nonuniform by number of particles (with content of spheres of each diameter equaled to 20% or 40% of total number of spheres) were generated. These idealized ternary mixtures represent interesting sets of spheres whose properties are scarcely known.

The porosity of samples with uniform PSD was found to decrease with an increase in the particle size ratio up to the certain critical particle size ratio above which the porosity remained unchanged. In spite of a decrease in the porosity of ternary mixtures, a few percentage decrease in the average coordination number was observed in the samples. Simultaneously, the corrected coordination number, including mechanically stable particles with more than three contacts, substantially increased with an increase in value of *g*. A study on the effect of the particle size ratio on the contribution of partial contact numbers into the global contact number and on the transmission of pressures through the contact network in ternary mixtures with PSD being uniform by number of particles has shown a presence of a critical *g* value above which the properties of packings remained unchanged. Analogous results have been found by authors for binary mixtures composed of spheres with different particle size ratios.

A study of the relationship between the fraction of particle size classes and the microstructure of ternary mixtures revealed a small effect of the content of particle fractions on the porosity and coordination numbers in packings, regardless of the *g* value. A strong influence of the number fraction of particle size classes on the partial stress in ternary mixtures was observed, which increased with an increase in the value of *g*. It was found that in ternary mixtures with small *g* value, the contribution of the partial stress into the global stress is determined mainly by number fraction of particle size classes, while it is determined by particle size ratio to a greater extent in packings with larger *g* values.

The findings of this study indicate some similarities, but also some differences between binary and ternary granular materials, which confirms a necessity for the investigation of the latter ones as thoroughly as mixtures comprising two components. A study on ternary materials with PSD uniform and nonuniform by the number of particles may lead to better understanding of real granular mixtures with more complex PSDs and more accurate prediction of effects that were observed in particulate assemblies.

## Figures and Tables

**Figure 1 materials-13-00339-f001:**
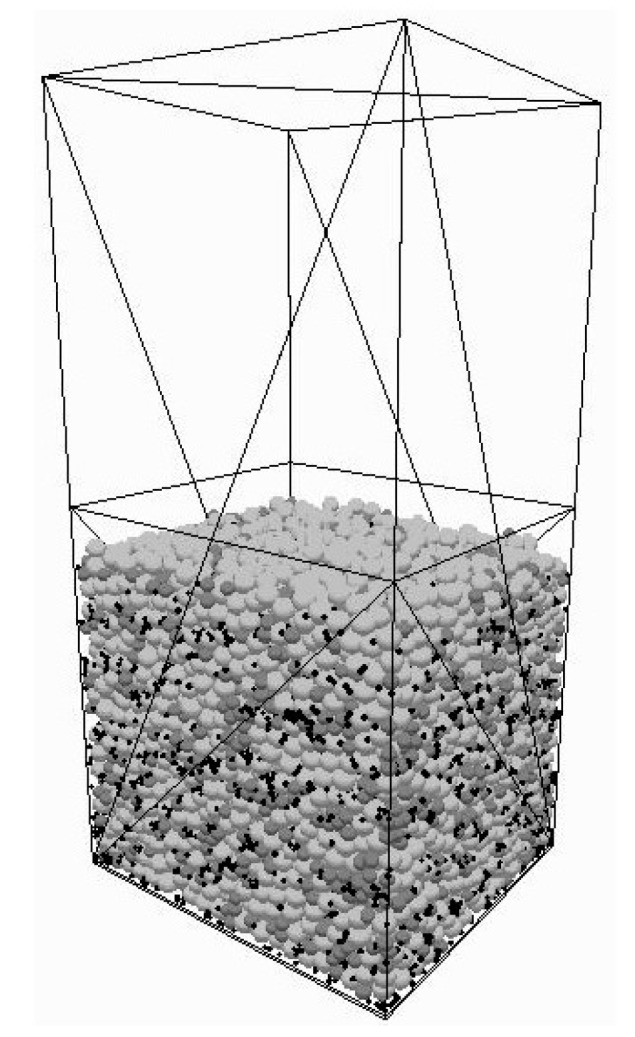
Initial configuration of mixture with size ratio of 3.75 and number fraction *f* = 424.

**Figure 2 materials-13-00339-f002:**
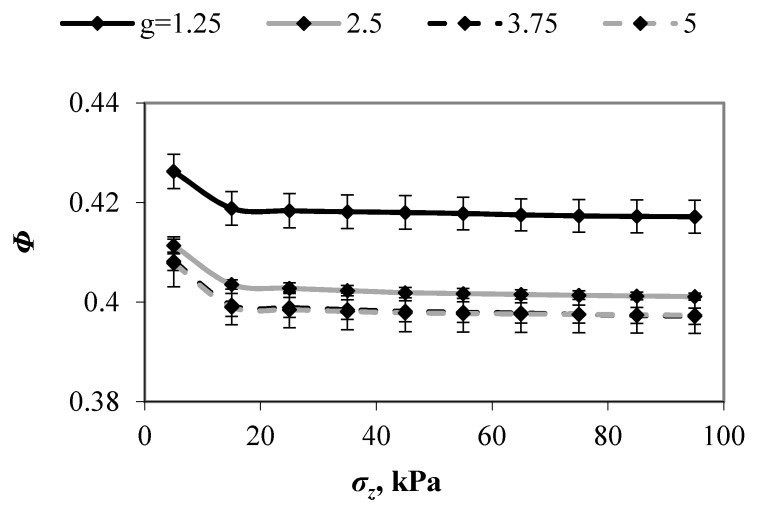
Evolution of porosity of mixtures with various values of *g* with imposed compressive pressures.

**Figure 3 materials-13-00339-f003:**
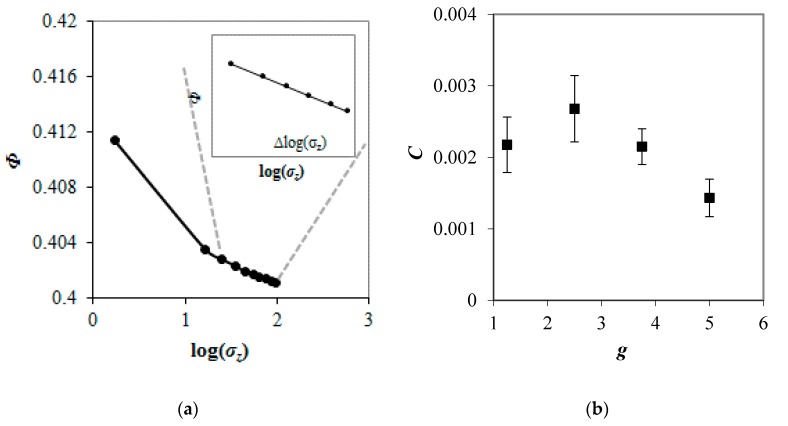
Relationship between the porosity and logarithm of vertical load for mixture with *g* = 2.5 (**a**) and evolution of compression index with particle size ratio in ternary mixtures under vertical load of 100 kPa; (**b**). Inset: zoom into the linear part of *Φ*(log(*σ_z_*)) curve.

**Figure 4 materials-13-00339-f004:**
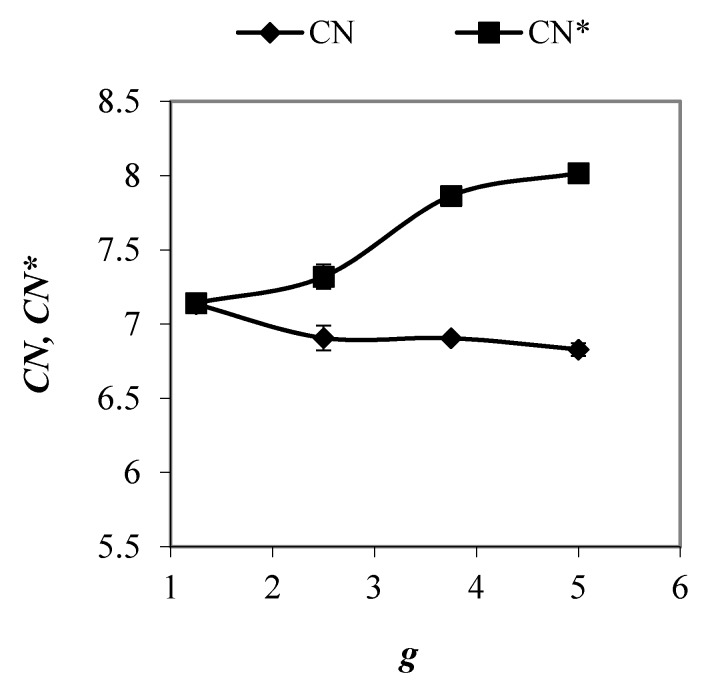
The average coordination number and the mechanical coordination number in mixtures with PSD uniform by number of spheres and various values of *g* when subjected to a vertical load of 100 kPa.

**Figure 5 materials-13-00339-f005:**
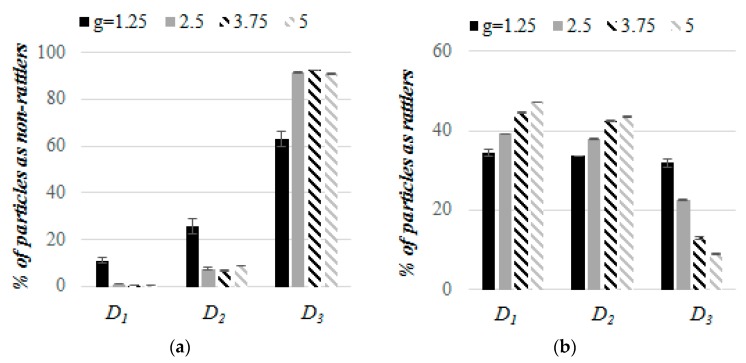
Percentage of the particles that are non-rattlers (**a**) and rattlers (**b**) in compressed mixtures with various values of *g* when subjected to a vertical load of 100 kPa.

**Figure 6 materials-13-00339-f006:**
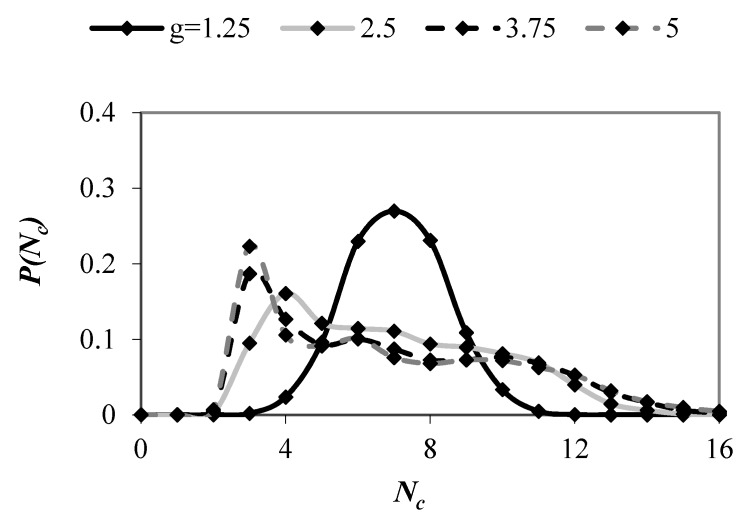
Probability distributions for the number of contacts per particle in mixtures with *PSD* uniform by number of spheres and various values of *g*.

**Figure 7 materials-13-00339-f007:**
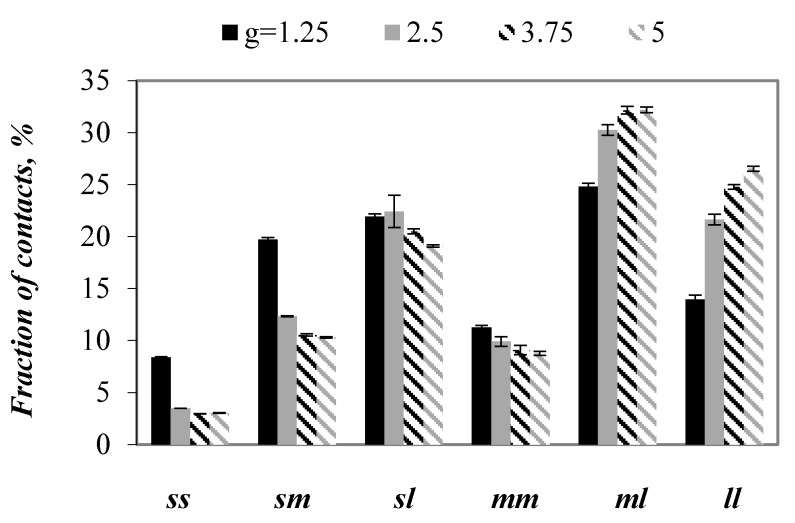
Percentage of number of contacts between small spheres (*ss*), small and medium spheres (*sm*), small and large spheres (*sl*), medium spheres (*mm*), medium and large spheres (*ml*), and large spheres (*ll*) in packings with various values of *g* when subjected to a vertical load of 100 kPa.

**Figure 8 materials-13-00339-f008:**
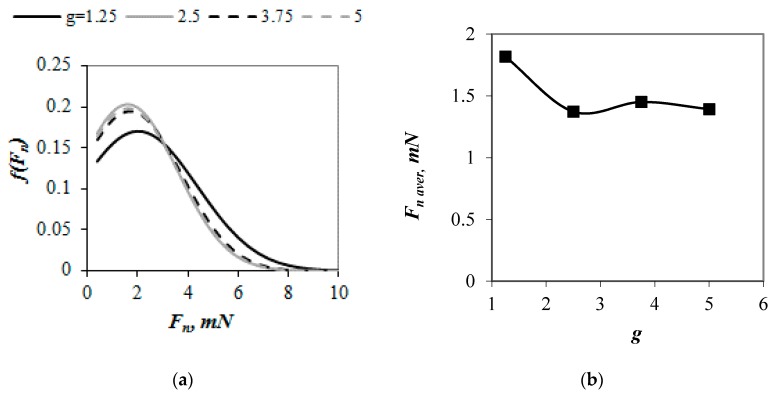
Average normal contact forces (**a**) and probability density functions of normal contact forces (**b**) in compressed mixtures with PSD uniform by number of spheres and various values of *g*.

**Figure 9 materials-13-00339-f009:**
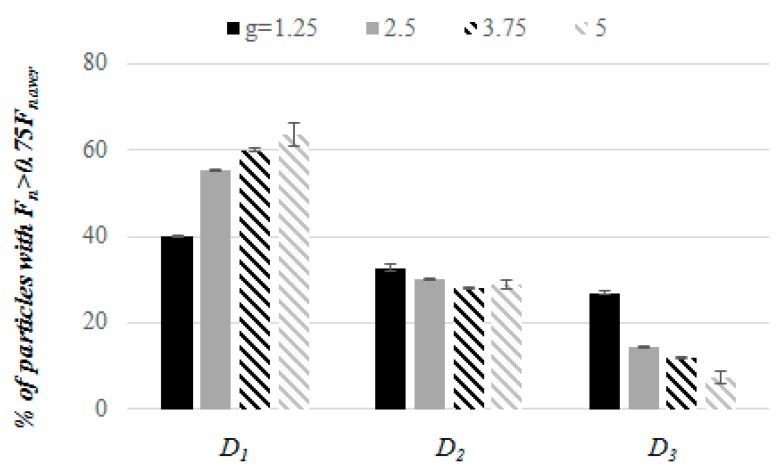
Percentage of contacts with *Fn* > 0.75 *Fn _aver_* in mixtures with various values of *g*.

**Figure 10 materials-13-00339-f010:**
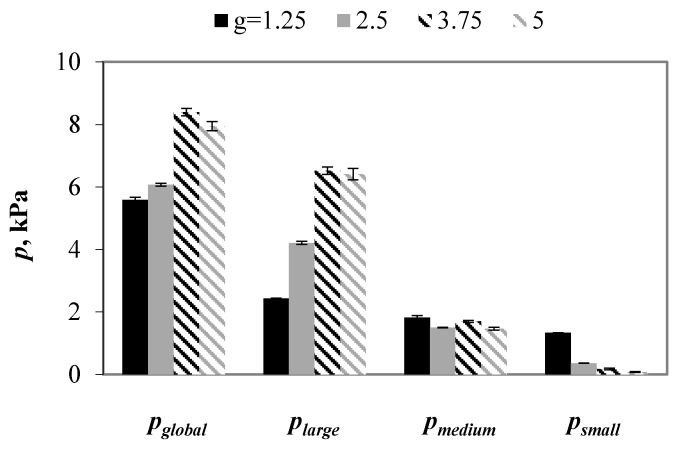
Global and partial stresses in mixtures with various values of *g* when subjected to a vertical load of 100 kPa.

**Figure 11 materials-13-00339-f011:**
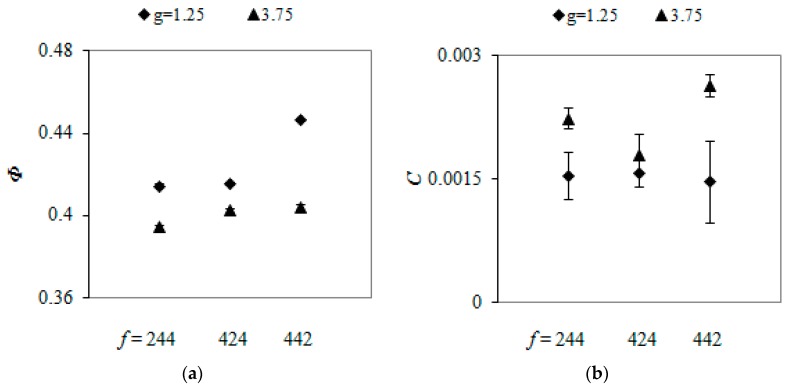
Evolution of porosity (**a**) and compression index (**b**) with number fraction of particles with different sizes in mixtures with *g* = 1.25 and *g* = 3.75.

**Figure 12 materials-13-00339-f012:**
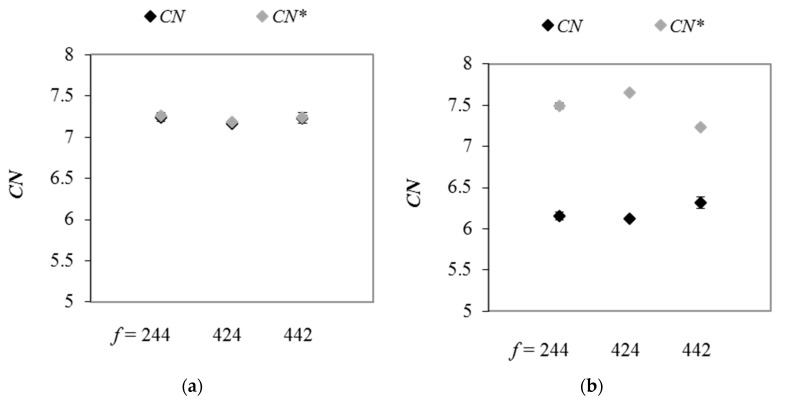
The average coordination numbers and mechanical coordination numbers in mixtures with *g* = 1.25 (**a**) and *g* = 3.75 (**b**) and various *f* values, under vertical load of 100 kPa.

**Figure 13 materials-13-00339-f013:**
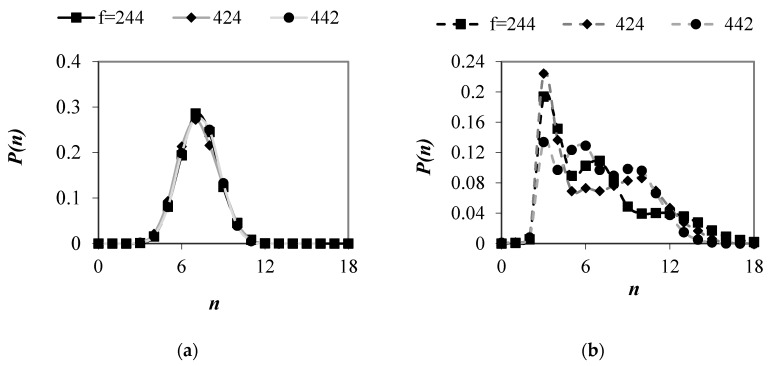
Probability distributions for the number of contacts per particle in mixtures with *g* = 1.25 (**a**) and *g* = 3.75 (**b**) and various f values, under vertical load of 100 kPa.

**Figure 14 materials-13-00339-f014:**
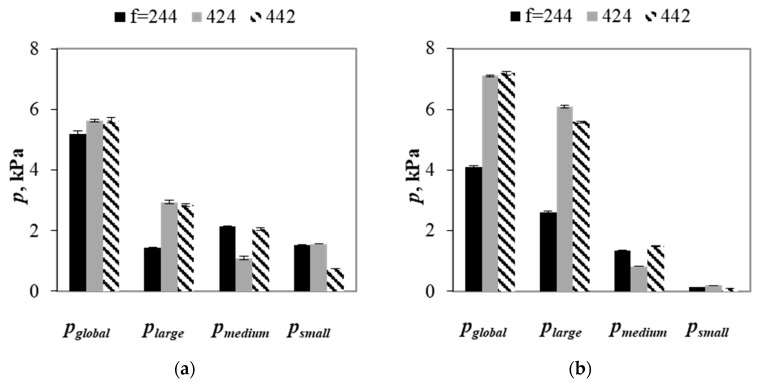
Global and partial stresses in mixtures with *g* = 1.25 (**a**) and *g* = 3.75 (**b**) and various values of *f*, under vertical load of 100 kPa.

**Table 1 materials-13-00339-t001:** DEM input parameters.

Parameter	Poisson’s Ratio *ν*	Shear Modulus *G*	Density *ρ*	Coefficient of Restitution *e*		Coefficient of Static Friction *µ_s_*		Coefficient of Rolling Friction *µ_r_*	
Value	0.3	77 GPa	7804 kg/m^3^	Particle-particle	0.4	Particle-particle	0.321	Particle-particle	0.01
				Particle-wall	0.4	Particle-wall	0.216	Particle-wall	0.01

**Table 2 materials-13-00339-t002:** Values of *D_k_* along with *N* for a given value of *g* in mixtures with particle size distribution (PSD) uniform by number of spheres.

*g*	*D_k_*, mm			*N*
	*D* _1_	*D* _2_	*D* _3_	
1.25	6	5.4	4.8	9450
2.5	6	4.2	2.4	13,800
3.75	6	3.8	1.6	18,000
5	6	3.6	1.2	18,300

**Table 3 materials-13-00339-t003:** Values of *f_k_* along with *N_k_* and *N* in mixtures with *g* = 1.25 and 3.75 and PSD nonuniform by number of spheres.

*g*	*f_k_*, %			*f*	*N_k_*			*N*
	*f* _1_	*f* _2_	*f* _3_		*N* _1_	*N* _2_	*N* _3_	
1.25	20	40	40	244	1890	3780	3780	9450
	40	20	40	424	3780	1890	3780	9450
	40	40	20	442	3780	3780	1890	9450
3.75	20	40	40	244	3000	6000	6000	15,000
	40	20	40	424	6000	3000	6000	15,000
	40	40	20	442	6000	6000	3000	15,000

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
