# Peer review of "Structural and Micromechanical Properties of Ternary Granular Packings: Effect of Particle Size Ratio and Number Fraction of Particle Size Classes"

_materials, 2020, doi:10.3390/ma13020339_

Round 1
Reviewer 1 Report
In this manuscript, the authors present the detailed studies on the effect of the particle size heterogeneity and the content of particle size classes on the structural and micromechanical properties of ternary particulate systems. These systems are commonly used in many branches of industry and the knowledge of their properties is very important. In this study, the relationship between the particle size ratio and the number fraction of particle size classes in ternary granular packings, and their structural and mechanical properties were investigated. A compression of mixtures composed of three particle size fractions was simulated with a discrete element method ( using the EDEM software).
In my opinion, the paper will be interesting to the readers. I think that the information in the article is properly arranged and the layout requirements of the Materials Journal have been met. Therefore, I recommend acceptance.
Author Response
Dear Reviewer 1,
I would kindly like to thank you for revision of the manuscript and for your acceptance for publishing it in the "Materials".
Yours sincerely,
Joanna Wiącek
Reviewer 2 Report
The article continually motivates and systematically leads the reader towards understanding the problem. The work sufficiently specifies the identified effects and gives possible recommendations. It does not deal with linearity/non-linearity in other possible scales (different material heights, size ratio particle to experimental box, effects of friction parameters) but solves micro-mechanical properties in general.
Reviewer 3 Report
This is a nice work of powder technology which is focused on the relationship between the particle size ratio and the number fraction of particle size classes in ternary granular packings, and their structural and mechanical properties. The results reported are interesting and original. The manuscript is well documented. Following points for discussion:
In the method part, it should be shown what governing equations were solved and in what procedure to deepen the reader's understanding. In the problem of compressive load shown in section 3.1.1, the amount of overlap between particles is an important issue and relates to the Young's modulus in DEM assumption. Can the authors consider that the volume conservation violation due to the amount of overlap between particles is small enough?Author Response
Please, see the attachment.

Reviewer 4 Report
The paper under review deals with the research on the structural and mechanical properties of ternary granular packings. The results in the study are achieved via modeling study (EDEM/DEM). The authors show models and results. It also includes the information on the available equipment and presents the research outcomes and their detailed description. A structure of the paper is in accordance with principles of scientific reports. The paper is written in good English. The article contains adequate and appropriately selected 39 literature items. In my opinion, the paper is acceptable for publication in Materials after minor revision.
Short comments:
Are there any experimental results to compare with the EDEM’s results? Please, add additional comment about practical aspects of the study (practical use of the results of the study). 2 – x-axis should be log. In my opinion the aim of the study is not clear (see comment No. 2).Author Response
Please, see the attachment.
